# Development of a 3D Perfused In Vitro System to Assess Proangiogenic Properties of Compounds

**DOI:** 10.3390/mps6060119

**Published:** 2023-12-09

**Authors:** Johanna Alm, Benoit Fischer, Alexandra Emanuela Burger, Francesca Moretti

**Affiliations:** Preclinical Safety, Biomedical Research, Novartis AG, CH-4056 Basel, Switzerland; johanna.alm@novartis.com (J.A.); benoit.fischer@novartis.com (B.F.); alexandra.burger@unibe.ch (A.E.B.)

**Keywords:** angiogenesis, vessel-on-chip, compound screening

## Abstract

Perturbation of angiogenesis is associated with a variety of diseases and pro- as well as antiangiogenic therapies are being actively explored. Additionally, unintended adverse drug effects on angiogenesis might lead to promotion of tumor progression and cardiovascular complications. Several tri-dimensional microfluidic vessel-on-chip systems have been described that allow a more accurate investigation of vascular physiology and pathology, compared to the two-dimensional static culture of endothelial cells. The OrganoPlate^®^ angiogenesis-on-chip system has been demonstrated to be amenable to high-throughput screening for the antiangiogenic properties of molecules. We set out to adapt this system for high-throughput screening of molecules with proangiogenic properties. Our technical advancement of the OrganoPlate^®^ angiogenesis-on-chip assay expands its applicability in the early screening of both anti- as well as proangiogenic properties of compounds for therapeutic modulation of angiogenesis as well as the identification of angiogenesis-associated drug-induced vascular toxicities.

## 1. Introduction

Angiogenesis is the process to form new blood vessels from a pre-existing vascular network. It is essential for many processes, such as embryo development and wound healing [1]. Abnormal angiogenesis (insufficient or excessive vessel formation) occurs in many diseases, such as cerebral stroke, heart disease, retinopathy, cancer, and chronic inflammation [2]. Therefore, proangiogenic agents have been explored as therapeutic options to treat peripheral vascular and coronary artery diseases, whereas antiangiogenic agents are being used to treat cancer, diabetic retinopathy, and macular degeneration, among others [3,4,5]. Given the central role that abnormal angiogenesis plays in the development of several diseases, it is crucial to ensure that misregulation of this process does not occur as a drug side effect. For example, cancer therapy with the tyrosine kinase inhibitor Nilotinib has been associated with the occurrence of progressive arterial occlusive disease [6]. Experimental evidence suggests that direct proatherogenic and antiangiogenic effects of Nilotinib on vascular endothelial cells may contribute to such adverse events [7]. GW501516 is a peroxisome proliferator-activated receptor (PPAR) β/δ ligand that was in development to treat dyslipidemia. The development of GW501516 was discontinued after observations of tumor induction in several organs in animal carcinogenicity studies [8]. It was demonstrated that activation of PPAR β/δ by GW501516 triggers endothelial cell proliferation and angiogenesis [9]. Successive work provided further evidence of the proangiogenic and tumor-promoting effect of PPAR β/δ agonism [10].

Several in vitro models of angiogenesis have been developed to study the sequential establishment and elongation of angiogenic sprouts as well as the direct compound effects on endothelial cell function [11,12]. Many of these assays are designed to assess compound effects on single processes contributing to angiogenesis, such as extracellular matrix (ECM) degradation and remodeling or cell proliferation and migration [13]. Until the advent of tri-dimensional (3D) microfluidic cell culture techniques, the ‘gold standard’ assay to assess the pro- and antiangiogenic properties of compounds was considered to be the tube formation assay [14]. In this assay, endothelial cells are grown on Matrigel or other ECM substrates and form tube-like structures. Major drawbacks of this assay are the long time needed (up to a few weeks) to establish the tubes and the absence of physiological perfusion and growth factor gradients [12]. The use of microfluidic cell culture techniques has fostered the development of in vitro systems that more closely mimic human vascular physiology and pathology [15]. While enabling the perfusion of the vascular tubes and the application of physiological growth factor gradients, several microfluidic systems to study angiogenesis require complex fabrication and provide only limited throughput [16,17]. Currently, the OrganoPlate^®^ angiogenesis-on-chip assay is one of the most advanced in vitro microfluidic models to study angiogenesis. This system allows the generation of tri-dimensional endothelial vessels with a perfused lumen and the stimulation of angiogenic sprouting with a growth factor gradient [18]. Contrary to other systems, it does not require sophisticated pumping systems, it is robust and scalable, as well as suitable for high-content imaging read-outs [19]. These characteristics make the assay appealing for medium/high-throughput screening applications. Published reports only describe the use of this assay to screen antiangiogenic compounds [20]. We describe here the adaptation of the OrganoPlate^®^ angiogenesis-on-chip protocol to enable the study of proangiogenic properties of compounds, with the goal of providing a unified platform to investigate drug-induced modulation of angiogenesis.

## 2. Materials and Methods

### 2.1. Cell Culture

Human Umbilical Vein Endothelial Cells (HUVECs) were acquired from Lonza (Basel, Switzerland, cat. no. CC-2517). Cells were grown at 37 °C and 5% CO_2_ in EBM-2 medium (Lonza, cat. no. CC-3156) supplemented with EGM-2 endothelial SingleQuots kit (Lonza, cat. no. CC-4176). Cells were passaged at 90% confluency and not used beyond passage 10.

### 2.2. Viability Assessment

In total, 20,000 cells/well were seeded in a 96-well plate. Then, 24 h after seeding, cells were treated with different doses of Astragaloside IV (0.1 to 30 µg/mL, MedChemExpress, Monmouth Junction, NJ, USA, cat. no. HY-N0431) and GW501516 (0.1 to 100 µM, Sigma Aldrich, Schnelldorf, Germany, cat. No. 317318-70-0) for 72 h. The compounds were obtained as powder and solubilized in 100% DMSO. Here, 0.1% DMSO (Sigma Aldrich, cat. No. D2438) was used as the experimental control, corresponding to the highest solvent concentration applied to compound-treated cells. Viability after treatment was assessed by measuring the intracellular ATP as a marker of metabolically active cells. The CellTiter-Glo^®^ 2.0 reagent (Promega, Düdendorf, Switzerland, cat. no. G9242) was used to measure intracellular ATP via oxidation of luciferin and emission of luminescence, following the manufacturer’s instructions.

### 2.3. Seeding in OrganoPlates^®^ and Angiogenesis Assay

HUVECs were seeded in the 3-lane OrganoPlates^®^ (Mimetas, Oegstgeest, The Netherlands, cat. no. 4004-400-B) according to manufacturer’s instructions [18]. Briefly, the top channel contained HUVECs, whereas the middle channel contained the extracellular matrix (ECM) gel. The gel was freshly prepared on ice by using 1 M HEPES (Gibco, Zürich, Switzerland, cat. no. 15630-080), 37 g/L NaHCO_3_ and 5 mg/mL collagen I (Cultrex Rat Collagen I, R&D Systems, Minneapolis, MN, USA, cat. no. 3440-100-01) in a ratio of 1:1:8; subsequently, 2 µL was dispensed in each chip. After gel polymerization (15–30 min at 37 °C; the polymerization time differed depending on the batch of the collagen I), 20.000 cells/chip were seeded in the top channel of each microfluidic chip. The OrganoPlate^®^ was incubated for at least 2 h on its side on the Mimetas plate stand to enable the cells to attach directly to the ECM. The plate was then transferred to the Mimetas OrganoFlow^®^ plate rocker (Mimetas, cat. no. MI-OFPR-S) with an inclination of 7° and a cycle time of 8 min. The cells were allowed to proliferate for 7 to 10 days, until a completely closed tube was observed. The angiogenic cocktail (or dilutions thereof as specified in the text) composed of 250 nM sphingosine-1-phosphate (Sigma Aldrich, cat. no. S9666), 37.5 ng/mL PMA (Sigma Aldrich, cat. no. P1585), 37.5 ng/mL human FGF-b (Peprotech, Cranbury, NJ, USA, cat. no. 100-18B), 37.5 ng/mL human MCP-1 (Peprotech, cat. no. AF-300-04), 37.5 ng/mL human HGF (Peprotech, cat. no. 100-39H), and 37.5 ng/mL human VEGF 165 (Peprotech, cat. no. 100-20) was added to the bottom channel of each microfluidic chip (Figure 1a). Where indicated, Astragaloside IV or GW501516 were added in the top channel and, additionally, together with the angiogenic cocktail in the bottom channel. Treatment durations ranged from 24 to 72 h. The compound treatment was refreshed every 24 h.

### 2.4. Nuclei and F-Actin Staining

Upon completion of treatment, cells were fixed with 4% PFA (Invitrogen, Zürich, Switzerland, cat. no. FB002) for 15 min, washed with PBS, permeabilized with 0.1% Triton X-100 (Sigma Aldrich, cat. no. 9002-93-1) for 10 min, and blocked with blocking solution (3% BSA in D-PBS, Invitrogen, cat. no. A14289SA) for 30 min. Nuclei and F-actin were stained for 90 min with Hoechst (Invitrogen, cat. no. H3570, 1:1000 in blocking solution) and Alexa Fluor 555 Phalloidin (Invitrogen, cat. no. A34055, 1:200 in blocking solution), respectively.

### 2.5. Image Acquisition and Analysis

The ImageXpress^®^ Micro Confocal high content imaging system from Molecular Devices (San Jose, CA, USA) was employed to capture images of the OrganoPlate^®^ chips. Briefly, Z-stacks of 40 fluorescence confocal images separated by 5 microns were acquired using a 10xPlan Apo Lambda objective, thereby enabling imaging of the full 200 µm depth of the chips. The resulting maximal projections of the 40 images of the Z-stack were subjected to image analysis using the MetaXpress^®^ software (version 6.7.2.290) from Molecular Devices.

#### 2.5.1. Sprouting Quantification

The sprouting quantification in the ECM gel was performed as follows: phase guide masks were first created applying an inclusive simple threshold on the maximal projection of Hoechst images, in a range of minimum and maximum intensities covering the autofluorescence intensity of the OrganoPlate^®^ phase guides; to clear small artefacts according to their size, we performed an erosion of the resulting mask using the ‘shrink objects’ feature with a value of 5 pixels; a filtering process based on the object area allowed us to reject any remaining small artefacts and keep only the mask of both phase guides. Since only the top phase guide mask was of interest at this stage, the bottom phase guide mask was excluded applying a filter based on the centroid Y coordinate of the objects; to fill the holes caused by heterogeneous autofluorescence or the presence of cell nuclei in the top phase guide mask, we applied a sequence of dilation followed by a shrink of the same value; the resulting mask overlapping the top phase guide of the chips was then inverted to create the two masks of the tube and ECM; we rejected the tubule mask, which was not of interest, by filtering it based on the centroid Y coordinate value, resulting in a mask that accurately corresponded to the ECM gel. The phalloidin-stained sprouts were detected in the ECM gel mask using the ‘Find Fibers’ algorithm. The resulting masks of fibrous objects and nonfibrous objects were merged to generate the mask of the sprout network. Sprout area, height, and length were calculated.

#### 2.5.2. Nuclei Centroid Y Sum Quantification

The detailed steps of this analysis are described in the results section. Briefly, in addition to the ECM gel mask, we applied a series of analysis steps to determine a reference line at the interphase between the endothelial tube and the ECM gel corresponding to the starting point from which the cells initiated migration into the gel. The ‘Find Round Objects’ algorithm was used on the Hoechst images to create a nuclei mask. We first measured the Y coordinate of all nuclei located in the ECM mask. To translate it into actual migration distances, we subtracted the Y coordinate of the reference top phase guide mask from the Y coordinates of the nuclei. The degree of angiogenesis was then analyzed calculating the sum of the migration distance of all nuclei.

## 3. Results and Discussion

### 3.1. Establishment of the OrganoPlate^®^ Angiogenesis-on-Chip Assay

We first set out to determine the optimized experimental conditions to allow the analysis of the proangiogenic properties of compounds in the OrganoPlate^®^ angiogenesis-on-chip assay. We seeded HUVECs in the upper channel of the OrganoPlate^®^ 3-lane against an ECM gel and allowed them to generate a perfused tube over 7 to 10 days. We then applied the established cocktail of angiogenic factors [18] in the bottom channel and analyzed angiogenic sprouting and growth over time (Figure 1a). We noticed that the application of the undiluted angiogenic cocktail (1:1 cocktail, as described in the protocol developed by Mimetas [18]), drives the establishment of a very dense sprouting network after both 24 (Figure 1b) and 48 h (Figure 1d), which would complicate the analysis of additive proangiogenic effects. This cocktail was indeed optimized for the analysis of antiangiogenic compounds [20]. Therefore, we set out to establish suboptimal sprouting conditions by means of diluting the angiogenic cocktail with cell culture medium and analyzing the sprouting at different time points (Figure 1b–e). The sprout area, height, and length were initially used to quantify the angiogenic phenotype, according to the published protocol [18]. We identified cocktail dilutions of 1:15 and 1:20 to trigger a substantial decrease in angiogenic sprouting, both at 24 and 48 h of treatment, and the sprout area as the parameter most strongly affected (Figure 1b–e).

### 3.2. Refinement of Image Analysis Pipeline

We noticed that quantification of the actin staining can be influenced by ECM stiffness and quality, which may vary between experiments. To increase the robustness of the assay, we therefore set out to establish a novel image quantification method that does not rely on actin staining but rather on the more reproducible nuclear stain. The position of the nuclei in the sprouts was already used to estimate the average sprout migration distance from the top phase guide in a previous study [20]. We refined this analysis by taking into account two parameters: 1. the position of the top phase guide needed to be determined for each field of view, as the chips were not always aligned; 2. we performed the nuclei-based quantification of sprout length by measuring the ‘Nuclei Centroid Y Sum’ as the sum of the distance of each nucleus in the gel channel from the phase guide. In this way, our analysis takes into consideration all nuclei present in sprouts rather than the 10 furthest migrating ones, as described previously [20]. Figure 2a shows the image analysis steps undertaken to determine the position of the top phase guide for every chip: we slightly dilated the top phase guide mask (step no. 1) to overlap it with the ECM gel mask (step no. 2). We then applied a logical operation “AND” between both masks resulting in a thin linear mask localized at the interphase of the tubule and the ECM gel (step no.3) and used this mask as the reference line from where the cells initiated migration into the gel. Instead of subtracting a constant value of 400 µm to the Y coordinates of the nuclei to evaluate the migration distance of nuclei as previously described [15], we rather subtracted the Y coordinate of the reference line, the location of which could differ from chip to chip. Angiogenesis was quantified calculating the sum of migration distance for all nuclei (Figure 2b). We applied this refined analysis method to the images obtained after 24 and 48 h of incubation with the angiogenic cocktail dilutions (Figure 1b–e) and observed comparable results (Figure 2c,d). We therefore used the refined analysis method to assess proangiogenic properties of the compounds.

### 3.3. Assessment of Proangiogenic Compounds

We selected two compounds to assess whether the protocol we had established was suitable to reveal both therapeutic as well as aberrant proangiogenic effects. Astragaloside IV is a natural product used in traditional Chinese medicine to treat cardiovascular disorders [21]. The therapeutic benefit of Astragaloside IV has been ascribed, among others, to stimulation of angiogenesis [22,23]. Development of the PPAR β/δ agonist GW501516 to treat dyslipidemia was discontinued due to tumor induction in animal studies [8]. Later studies revealed tumor-promoting as well as proangiogenic properties of GW50156 [10]. Astragaloside IV is not cytotoxic to HUVEC cells up to 30 µg/mL over 72 h of treatment; GW50156 is not cytotoxic to HUVEC cells up to 10 µM over 72 h of treatment (Figure 3a). In the first angiogenesis experiment, we incubated two doses of the selected compounds in both the vascular tube lumen and in the third channel of the chip together with the diluted angiogenic cocktail (1:15 and 1:20 dilutions) for 48 h (Figure 3b). We observed a two- to fourfold increase in angiogenic sprouting with both doses of Astragaloside IV in presence of either 1:15 or 1:20 diluted angiogenic cocktail (Figure 3b,c). The ≈twofold pro-angiogenic effect of GW501516 was best observable in the presence of the 1:20 cocktail (Figure 3b,c). The effect in the presence of the 1:15 cocktail was masked by a high chip-to-chip variability in the assay (Figure 3c). The proangiogenic effects of Astragaloside IV and GW501516 were confirmed in two additional independent experiments.

To explore whether a longer incubation time would improve the outcome of the assay, we extended the treatment to 72 h (Figure 4). Compared to the 48 h treatment (Figure 3b,c), 10 µg/mL Astragaloside IV showed very comparable results when incubated with the 1:15 cocktail but a lower proangiogenic effect in presence of the 1:20 cocktail (Figure 4). GW501516 displayed a stronger proangiogenic effect at 72 h only at the lower tested dose and in presence of the 1:15 cocktail (Figure 4). All the other conditions yielded worse results compared to the 48 h incubation (Figure 3c and Figure 4b). It is reasonable to hypothesize that the different outcomes of the assay for the two compounds tested might be due to different molecular mechanisms underlying the proangiogenic effect.

Despite obtaining convincing results, we did notice chip-to-chip variability and therefore encourage end-users of this protocol to account for at least four technical replicates per condition. Variability in sprouting densities were also observed in a previous study and believed to be caused by differences in cell seeding densities [20]. Furthermore, this protocol has been adapted to HUVECs. Adaptation to other endothelial cell types (and even different lots of HUVECs) might require optimization of the angiogenic cocktail dilution and treatment duration as we did for HUVECs. Finally, assay conditions and treatment durations might need to be optimized for individual compounds, according to the molecular mechanism of angiogenesis stimulation.

## 4. Conclusions

In this work, we have expanded the applicability of the OrganoPlate^®^ angiogenesis-on-chip to study the proangiogenic properties of compounds. We here provide a detailed protocol and a refined image analysis pipeline that enabled us to detect the described proangiogenic effects of two compounds, Astragaloside IV and GW501516. This assay now allows standardized medium-/high-throughput screening of positive and negative modulators of angiogenesis for therapeutic intervention and identification of vascular toxicities.

## Figures and Tables

**Figure 1 mps-06-00119-f001:**
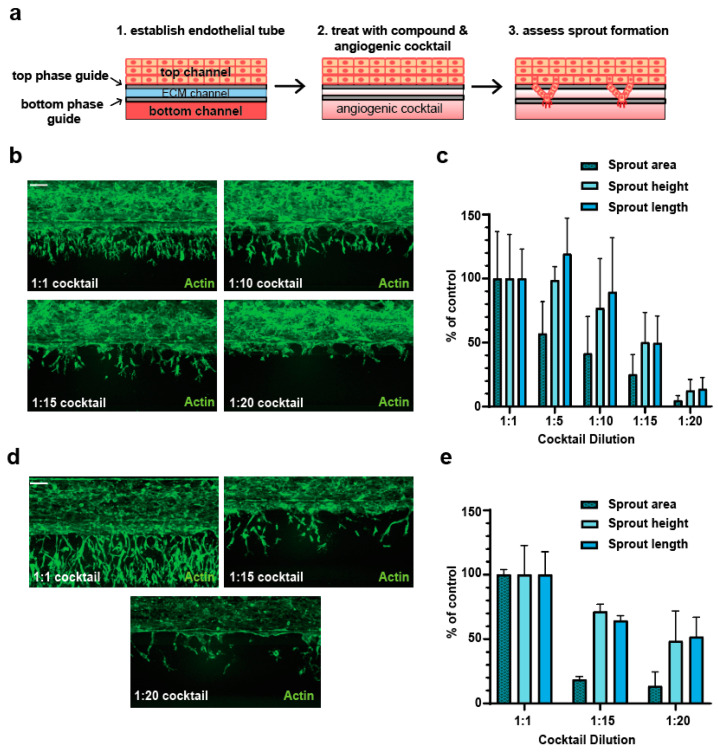
(**a**) Schematics of the OrganoPlate^®^ angiogenesis-on-chip assay. (**b**–**d**) The endothelial tubes were treated for 24 (**b**,**c**) or 48 (**d**,**e**) hours with different dilutions of the angiogenic cocktail, fixed and stained with Phalloidin. Images were acquired with the ImageXpress MicroConfocal microscope, and maximal projections were analyzed with the MetaXpress software. Sprout area, height, and length were quantified for the cocktail dilutions using undiluted cocktail as the control. Plotted is the average and standard deviation (n = 3). Scale bar = 100 µM.

**Figure 2 mps-06-00119-f002:**
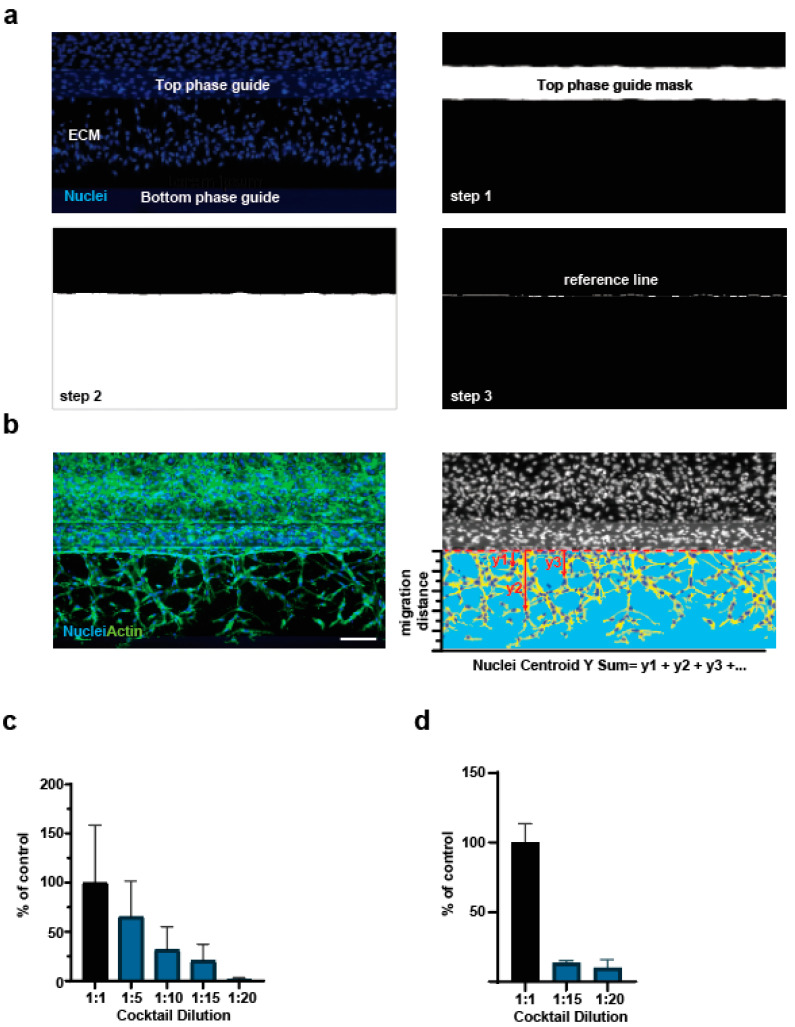
(**a**) Example images of the analysis steps necessary to determine the position of the reference line. (**b**) Example images of nuclei identification and quantification of centroid Y Sum. The red dotted line corresponds to the reference line as determined with the steps presented in panel (**a**). Scale bar = 100 µM. (**c**,**d**) Quantification of the nuclei Y centroid sum of angiogenic sprouting after 24 (**c**) or 48 (**d**) hours incubation with the angiogenic cocktail. The undiluted cocktail is used as control. Plotted is the average and standard deviation (n = 3).

**Figure 3 mps-06-00119-f003:**
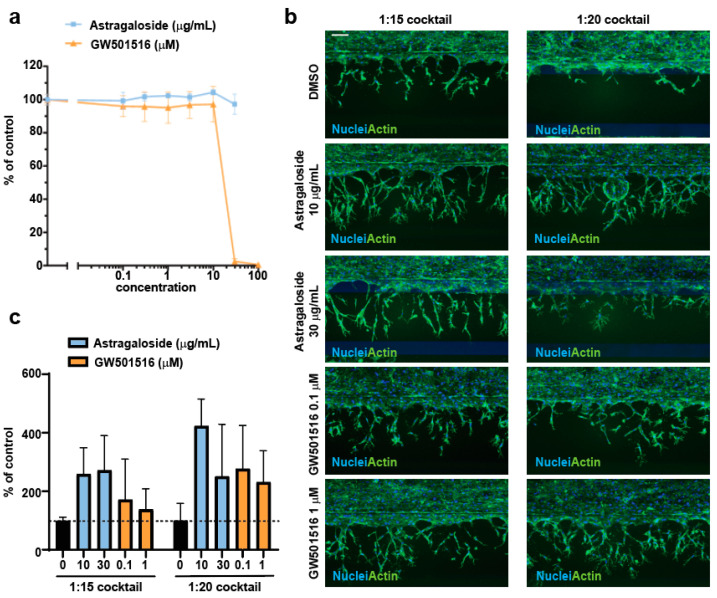
(**a**) HUVECs were seeded in 96-well plates and treated with the indicated compounds for 72 h. Viability was assessed as % to DMSO control. Plotted is the average and standard deviation (n = 3). (**b**) The endothelial tubes were treated with the indicated compounds for 48 h. After treatment, cells were fixed and co-stained with Hoechst and Phalloidin. Scale bar = 100 µM. (**c**) Sum of nuclei Y centroid was calculated as % to DMSO control. Plotted is the average and standard deviation (n = 3–4).

**Figure 4 mps-06-00119-f004:**
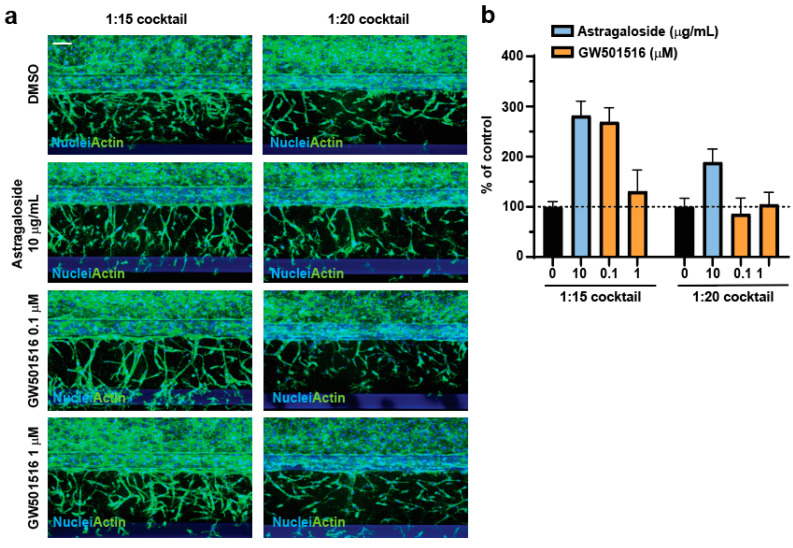
(**a**) The endothelial tubes were treated with the indicated compounds for 72 h. At the end of the treatment, cells were fixed and co-stained with Hoechst and Phalloidin. Scale bar = 100 µM. (**b**) The sum of the nuclei centroid was calculated as % to DMSO control. Plotted is the average and standard deviation (n = 4).

## Data Availability

The data presented in this study are available on request from the corresponding author. The data are not publicly available due to restrictions.

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
