# Peer review of "Development of a 3D Perfused In Vitro System to Assess Proangiogenic Properties of Compounds"

_mps, 2023, doi:10.3390/mps6060119_

Round 1

Reviewer 1 Report

Comments and Suggestions for Authors

Reviewer 2 Report

Comments and Suggestions for Authors

This interesting work provides an option for evaluating the pro-angiogenic of materials. This paper was well-organized with high novelty. Before acceptance, I suggest the authors should: 1) further improve Figure 1a according to the text, so the readers can clearly understand the OrganoPlates system; 2) In the introduction, some new angiogenesis evaluation methods can be cited, such as https://doi.org/10.1016/j.ijbiomac.2023.125201 

Reviewer 3 Report

Comments and Suggestions for Authors

During the methodology presented in the paper, I noticed that you cited many protocols such as Mimetas (13) without briefly describing how they were used.

As this is a journal of protocols and methods, and your study has interesting results and new techniques, I suggest you be more descriptive in the materials and methods section. 

item 2.3 line 74

The cells were allowed to proliferate for 7 to 10 days, until a completely closed tube was observed.

What time period was used?

item 2.4, lines 91 to 93

Nuclei and F-actin were stained for 90 minutes with Hoechst (Invitrogen, cat. no. H3570, 1:1000 in blocking solution) and Alexa Fluor 555 Phalloidin (Invitrogen, cat. no. A34055, 1:200 in blocking solution) respectively.

Only one time was quoted for F-actin, and Phalloidin? how was it done? after blocking with f-actin, did you wash and incubate with another one afterwards or did you use 2 independent assays? 
